# Perspectives on Scaffold Designs with Roles in Liver Cell Asymmetry and Medical and Industrial Applications by Using a New Type of Specialized 3D Bioprinter

**DOI:** 10.3390/ijms241914722

**Published:** 2023-09-29

**Authors:** Iuliana Harbuz, Daniel Dumitru Banciu, Rodica David, Cristina Cercel, Octavian Cotîrță, Bogdan Marius Ciurea, Sorin Mihai Radu, Stela Dinescu, Sorin Ion Jinga, Adela Banciu

**Affiliations:** 1Department of Biomaterials and Medical Devices, Faculty of Medical Engineering, Politehnica University of Bucharest, 1-7 Gh. Polizu Street, 011061 Bucharest, Romania; iuliana.harbuz@upb.ro (I.H.); octavian.cotirta@stud.fim.upb.ro (O.C.); bogdan.ciurea@upb.ro (B.M.C.); sorin.jinga@upb.ro (S.I.J.); 2Institute for Research on the Quality of Society and the Sciences of Education, University Constantin Brancusi of Targu Jiu, Republicii 1, 210185 Targu Jiu, Romania; rodica.david@e-ucb.ro; 3Department of Mechanical Industrial and Transportation Engineering, University of Petrosani, 332006 Petrosani, Romania; sorinradu@upet.ro (S.M.R.); steladinescu@upet.ro (S.D.); 4University of Medicine and Pharmacy “Carol Davila” Bucharest, 37 Dionisie Lupu Street, 020021 Bucharest, Romania; cristina.cercel15@gmail.com

**Keywords:** cellular asymmetry, scaffold, 3D printing

## Abstract

Cellular asymmetry is an important element of efficiency in the compartmentalization of intracellular chemical reactions that ensure efficient tissue function. Improving the current 3D printing methods by using cellular asymmetry is essential in producing complex tissues and organs such as the liver. The use of cell spots containing at least two cells and basement membrane-like bio support materials allows cells to be tethered at two points on the basement membrane and with another cell in order to maintain cell asymmetry. Our model is a new type of 3D bioprinter that uses oriented multicellular complexes with cellular asymmetry. This novel approach is necessary to replace the sequential and slow processes of organogenesis with rapid methods of growth and 3D organ printing. The use of the extracellular matrix in the process of bioprinting with cells allows one to preserve the cellular asymmetry in the 3D printing process and thus preserve the compartmentalization of biological processes and metabolic efficiency.

## 1. Introduction

The field of 3D printing has undergone extensive development in recent years [1]. The number of published scientific articles is growing exponentially in this field, but this growth has not been accompanied by an increase in commercially available technologies [2]. In the medical field, there is an increasingly acute demand for the development of 3D printing technologies that can work with biological systems [3,4]. Because there is a significant number of patients waiting for an organ donor but very few donors [5,6,7], since 2018, several attempts have been made to create organs using 3D bioprinting [8].

The first technologies developed for this purpose were based on the use of spheroids [9] and the assumption that cells behave uniformly and self-organize locally based on environmental factors and intrinsic genetic factors [10]. Through using such technologies, blood vessels [11] and muscle-like structures [12] have been created, and these structures have been shown to organize themselves according to factors such as mechanical strength and proximity [13].

Although significant amounts of money have been invested in the research and development of 3D printing methods [14,15,16,17,18,19,20,21,22], they have failed to develop complex tissue systems [23] such as kidneys [24] and livers [25], as they started from a simplified assumption of cellular functioning. This simplification was imposed by technological limitations [26] such as the inability of 3D printing technologies to use large clumps of cells, as well as the willfully ignorant communication within this limited environment [27]. Going beyond the current boundaries of 3D tissue printing requires a degree of interdisciplinarity that stems from biomedical knowledge and makes use of technological knowledge and not the other way around (as has been the case so far).

## 2. Various Methods of 3D Bioprinting and Their Relation with Cell Asymmetry

In order to obtain a 3D cell structure that is already oriented with respect to cellular asymmetry directly from the printing phase, a few criteria must be met: a resolution smaller than cell size, sequential printing (without continuity elements that block cellular orientation), and no toxic factors (such as UV radiation or high temperatures). The current methods of 3D bioprinting are presented in Table 1 (ordered according to the above criteria) [28].

Structures that do not require special technologies to guide cellular asymmetry (e.g., blood vessels) can be printed using classical techniques [29].

It is necessary to identify the functional requirements of the tissues and the effect of each variable on said tissue [30] in order to provide engineers with all of the necessary data so that they can make the best use of the current technological possibilities.

Cell communication is carried out on different levels: through direct interaction between cells, at a distance through soluble factors [31,32,33], and via matrix cell communication. These three main types of communication are the main factors that must be considered in 3D bioprinting, thus providing the following rules for a successful bioprint:The 3D printing must have a resolution of one cell.The 3D printing must allow for real-time remote communication.The 3D printing must allow for cell–cell and cell–matrix communication at the same time.

These factors have a few consequences. The cultivation of cells before 3D printing should allow them to function physiologically at the target site. This first aspect is important from the point of view of cellular asymmetry. Certain cell types have an anisotropic distribution of cell content depending on the aforementioned communication factors [34,35]. For example, cells from the intestinal absorption mucosa and cells from the hepatic parenchyma and the renal parenchyma, respectively, have the transport of substances directed according to the signaling factors that change the distribution of cell transport factors [36]. This communication is two-way communication. At the basal pole, the cells communicate with the extracellular matrix in the basal membrane, and at the lateral pole, the cells communicate with neighboring cells.

## 3. Cellular Spot

The smaller the size of the cellular spot, the more accurate the 3D printing can be. The larger the size, the better the cell anisotropy is preserved. The size of the cellular spot attached to the basal membrane cannot exceed a certain size due to spatial limitations. Reorganizing spot structures around blood vessels requires a degree of curvature that is different from reorganizing into spots for intestinal absorption epithelia.

The size of the cellular spot not only changes the 3D printing conditions but also affects the ability of the cells to self-organize. This unicellular spot must be relatively mechanically stable in the printing process. To preserve cell anisotropy, the cellular spot must already be organized before printing. A two-dimensional cell culture on an extracellular matrix substrate is required to guide its anisotropy. This substrate must be able to be successfully identified by cells but must also meet other conditions. The most obvious of these is the ease of fragmentation of the single-celled layer. This fragmentation can be achieved during or after the break-up. For maximum efficiency, organizing a cytoskeleton of a cellular confluence may be required depending on the anisotropy criteria. Due to the difficulties in predicting two-dimensional cell growth [37], a high degree of freedom is required, which can be obtained by preserving the substrate that mimics the basal membrane.

A consequence of the two-dimensional uniformity of the basal membrane is represented by the need to identify the positions of the adjacent cells. Cutting the cell support and catapulting [38] is a method used in clinics and research for single-cell analysis. Unlike current methods that use an artificial membrane that facilitates separation, in order to facilitate the 3D printing of cellular spots, it is necessary to avoid any foreign plastic material. For this purpose, the realization of a support membrane is required, which can be broken down in the catapulting process. Among the current approaches, several variants that can meet these criteria can be identified, the simplest of which being the layering of the cellular support. The lower layer will be used for the detachment and catapulting phases and will subsequently dissolve, while the upper layer provides the basal membrane function. Subjecting the layer to laser-stimulated evaporation, which facilitates separation and catapulting, can be achieved by using materials that dissolve in a physiological environment but whose dissolution rate is reduced in a way that allows for the basal membrane to be added. Combinations of substances with different rates of dissolution in water, such as bio cellulose or polyvinyl alcohol, may be used to allow for the adjustment of the rate of dissolution by changing the ratio of the two components.

If one would like to design a biomaterial, they must consider the following properties: biocompatibility, biodegradability, and bioabsorbability [39]. Other related factors, such as porous structures which allow for good cellular adhesion and the ability to promote cellular interaction, must be considered to eventually achieve the development of a tissue. Also, the mechanical properties influence the viability of a biomaterial due to its resistance to wear. Bacterial Cellulose (BC) and composite materials with BC have already been used for wound dressing purposes, curing burns, and developing artificial skin. Recently, there have been some attempts to obtain artificial organs [40] using BC.

BC has properties that indicate that it can be used in biomedical applications, such as its good crystallinity, network structure, and 3D nanofibril network, all of which enhance its mechanical properties. Also, due to its chemical nature and the nanofibrillar architecture, BC itself looks like an extracellular matrix. Moreover, the hydrophilic nature of BC gives it a high water holding ability and adhesion [41].

In some biomedical applications, such as skin treatment, wound healing [42], and bone scaffolds [43], bio absorbability is not a desirable property. In these cases, BC is appropriate for use because these applications require long-term support [44].

Because the human body is not able to degrade cellulose due to the absence of cellulases, various attempts to make BC bioabsorbable in order to increase the applicability of BC have been made through the oxidation of cellulose or by incorporating enzymes into BC hydrogel.

Polyvinyl alcohol (PVA) is used as a component to obtain biomaterials because of its properties (small-molecule permeability, low interfacial tension, soft consistency, transparency). The mechanical characteristics of a PVA can be enhanced based on the nano-fibrous structure of BC and the formation of hydrogen bonds between BC and PVA [26,45,46]. Importantly, BC hydrogels have high bioactivity, promoting the migration of cells, which leads to the acceleration of tissue formation [40,45,46].

The adhesion of the support layer to the culture vessel can be achieved in the classic variant or by a non-chemical, electrostatic, or magnetic-mediated adhesion to avoid mechanical stress in the catapulting process (Figure 1). In this case, the support membrane is attached or detached from the substrate by the electric or magnetic field in which the culture vessel is located to the existence of electrically charged or magnetizable elements (such as iron oxide) in the basal membrane. Ideally, these components can be removed by, for example, dissolving a layer of the support membrane.

The laser cutting of the support substrate and monolayer can be carried out completely before any catapulting, or they can be done simultaneously. In both cases, it is necessary to remove the sectioned cell fragments.

Removing cellular debris before catapulting involves breaking the links between the integrins of fragmented cells (in the process of laser cutting) to the substrate [47,48] while keeping the connections between the integrins of living cells and the substrate of extracellular matrix intact. This selectivity step can be achieved via intracellular mechanisms to adjust the integrins’ adhesion to the extracellular matrix [49,50]. The introduction of intracellular signaling factors of release into the culture medium and to the substrate only influences the internal faces of the integrins in the cells with compromised cellular integrity, i.e., those affected by the laser cuts.

At the same time, it is necessary to release the links made between the cadherins between the cell fragments and viable cells [51], but it is critical that the connections between viable cells remain untouched [52,53]. If this intercellular detachment is achieved after the fragmentation of the debris under the extracellular matrix layer, agitation and mechanics of the culture medium (such as orbital stirrer or crop medium jets) place a higher degree of mechanical tension on the links between the cadherins that connect cellular debris to intact cells compared to the cadherins that bind living cells. The additional use of a calcium chelator leads to the relatively specific unbinding of these connections between living debris and living cells. After the rapid removal of cellular debris, the calcium chelator is also removed, which allows for the eventual restoration of the links between the cadherins between adjacent living cells if they have been denatured due to the cellular proximity preserved by the adhesion to the substrate. The removal of cellular debris can be monitored under a microscope.

The removal of cellular debris after catapulting involves vortex-type agitation [52,54]. This stage requires a thick support membrane in order to not put mechanical tension on the bonds between living cells. When the thickness of the support membrane is close to the thickness of a cell, its partial dissolution will be more prominent toward the periphery, i.e., toward the cell fragments (especially if the cellular spot size is only two cells). Cellular debris removal can be monitored diffractometrically [55] via relatively slow changes in the dissolution of the support material (only the lower part; not the similar extracellular matrix) and relatively fast changes in the release of membrane debris (from the laser cut cells). In this case, calcium chelators can be used for a short time depending on the slopes of the growth rate of light diffraction angles (correlated with the appearance of small free membrane fragments and with the decrease in the size of cellular spots), with cellular separation and resuspension in the calcium-free environment, or calcium overload of the environment if centrifugation is avoided (for the preservation of cellular spots). This stage of removing cellular debris after catapulting from the substrate is more effective at large volumes of cells compared to the method of debris separation before catapulting; however, it requires laborious steps to adjust the rate of substrate dissolution and quantify the release of cellular debris.

## 4. Positioning and Orientation in the 3D Printing Process

Unlike usual bioprinters, 3D printing that considers cellular anisotropy requires a high degree of accuracy with respect to the three classic X, Y, and Z axes [56], as well as with respect to the orientation of the cellular spot on three rotational axes. The classic 3D positioning limits are complicated by the need to investigate the rotation degree to modify these rotation axes. In order to rotate the cellular spot electrical forces, magnetic forces, and optical forces [57,58] can be used. In either variant, the application of the rotational force must cease once the 3D printing material touches the support. Due to the relatively high degree of non-uniformity among the cellular spots, the simplest assessment method investigates the support matrix using optical or fluorescence methods.

The use of electrical forces to rotate [59] the cellular spot requires the asymmetric electric charging of the spot and the neutralization of the electric charges when the tissue that is being 3D printed touches the support. The simplest method for this involves selecting electrical charges based on the pH of the solution compared to the isoelectric pH [60]. A direct consequence of this approach is the need for the 3D piece to be infused in order to normalize the pH and neutralize the electric charges capable of further rotating the cellular spot after the deposition of the tissue that is being 3D printed. The ability to apply mechanical forces to the electrically charged spot confers the potential advantage of directing the cellular spot with increased precision.

The use of mechanical forces secondary to magnetic interactions to rotate the 3D spot is difficult to use due to the difficulties in magnetic focus and difficulties in building a cellular support that loses its magnetic capabilities when touching the 3D piece.

The optical tweezer allows for the application of mechanical forces, including rotation. These forces are in the order of pico-newtons and require the laser spot to overlap with the cellular spot particle. The holographic rotation pattern requires a predictable extracellular environment; thus, it is necessary to ensure a constant velocity movement (preferably within a transparent capillary).

Due to the aforementioned characteristics of the magnetic and optical manipulations, the simplest constructive variant for ensuring the orientation of the cellular spot is the electric one.

## 5. The 3D Printing of Cellular Tissue

The implementation of positioning and rotation movements in the 3D printing process based on electrostatic forces can be achieved sequentially by considering the differences between rotational inertia and displacement inertia.

In order to position the cell spot, a technique similar to a flow cytometric sorting process is used. The cell spot is suspended in a drop of liquid of variable size. The droplet size correlates with the moving forces in the electric field but does not correlate with the friction forces between the cellular spot and the liquid in the drop. By manipulating the droplet size, the differences between the rotational and moving forces can be accentuated to allow for sequential rotation and movement (initially the movement that has higher inertia and then the one with low inertia). The two types of mechanical inertia (moving on sliding or rotational axes) should be evaluated at the level of each drop in the 3D printing process (Figure 2A). In this regard, carrying out position assessment via optical methods is required. Colorimetric 3D reconstruction variants using multiple cameras can record the position of the support material relative to the drop volume (correlated with the drop position) with or without the use of fluorescent images (Figure 2B). In the latter case, the diffraction of light provided by the liquid droplet must be considered, which may complicate the analysis of the rotation of the cellular spot but could increase the rate of evaluation if several fluorophores are being evaluated simultaneously (through multiple cameras associated with dichroic cascade mirrors).

Positioning cellular spots made up of cells and support material is not enough for the tissue to function as a whole. It is necessary to obtain the adhesion forces between different cellular spots and to achieve the permeability of the microscopic tubes while also achieving impermeability between them. The simplest method of achieving impermeability is adherence between various support matrices and cell element gaps, so that intercellular spaces on one side of the fusion matrix are offset by intercellular spaces on the other support matrix with which it comes in close contact. For this contact to remain tight, we need an additional reticulation process or a similar connection process between various support matrices (each one specific to its own cellular type) and without homophile self-assemblies. These criteria can be fulfilled by producing fibers from the extracellular matrix via an electrospinning process that includes contralateral matrix antibodies. Thus, over a sufficiently long period of time, we can achieve the stability of the three-dimensional structures to allow our cells to synthesize the extracellular matrix, which ensures the continuity of cellular structures and for cadherins to seal intercellular spaces. Antibody antigen reactions (antigen on one support network; antibody on another contralateral support network) may take time to stabilize the 3D construction piece, and this time can be ensured via chemical reticulation processes (preferably using ions other than calcium).

The preservation of vascular (and not only) canalicular structures [61] may be maintained via permanent infusion [62]. This ensures cellular viability, especially in large 3D constructions and when neutralizing the electrical charges of the support matrix (which was used to rotate the cellular spot) to prevent the application of electrical forces to the cells after they reach the 3D construction tissue.

## 6. Printing with Different Cell Types

For large tissue that is being 3D printed with different cell types, the accurate localization of the cell type and a high printing speed are required. Apparently, the two variables are contradictory, but the ability to spatially guide the distribution of the cellular spot in the electric field allows for the fine adjustment of the position of the cellular spot in relation to the printing support. This is more common in highly repetitive structures such as liver parenchyma. In this regard, a cellular sorter (of cellular spots) that has the ability to guide multiple cellular inputs and orientate them depending on the model built can be built. This technological solution has as the advantage of the relatively high speed of 3D printing, but its main disadvantage lies in the significant mechanical impact it has on the cellular spots and the 3D piece, which can change cellular orientation and, as a result, cellular anisotropy. In addition, such mechanical forces [63] can lead to the differentiation of the stem cells [64] used in the printing process, which must be compensated quantitatively. In order to reduce this risk of impact on rotation, the cell construction is continuously infused, thus obtaining uniform elasticity.

## 7. Printing of a Liver Tissue with Functional Cellular Asymmetry

Previously, In the US, the number of patients awaiting a liver transplant (LT) was about 11,844, of which 8250 received a transplant [65]. However, when the COVID19 pandemic began in 2019, the number of living donors for liver transplants decreased by 49%, while the number of deceased donor liver transplants reduced by 9% [66]. The medical costs for patients on the waiting list vary dramatically in terms of the MELD (Model for End-Stage Liver Disease) score. In the first year of diagnosis, the treatment can cost approximately USD 49,407 for the patients with the lowest MELD scores and about USD 613,020 for those with the highest MELD scores. In the primary year after receiving a LT, the cost can fluctuate between USD 588,850–USD 836,788 [67]. Since the medical costs are greater for patients with high MELD scores, offering an increased number of LTs to patients with low MELD scores would be profitable for the economy.

The liver has a highly repetitive structure that allows for the use of the described technology at a relative velocity as a cellular assortment via flow cytometry (between 30,000 and 100,000 distinct elements per second) [68]. Starting from an average volume of hepatocytes of 3.4 × 10^−9^ cm^3^ and from the use of cellular spots consisting of only two cells, we need about 1.7 × 10^9^ sorting elements per cubic centimeter of liver, that is, 17,000 s (at the speed of 100,000 per second), i.e., about 5 h. Considering that our construction is permanently infused, we do not have a time limit; only economic ones. We need at least five cell types, namely, hepatocytes, endothelial cells, fibroblasts [42], smooth muscle cells, stem cells [69]), of which only hepatocytes and endothelial cells have a high degree of anisotropy (guided by the basal membranes and also synthesized by the fibroblasts). To ensure a high degree of mechanical stability, increasing the diameter of the capillaries and bile ducts can help to prevent a sharp liver repair (like the process of liver cirrhosis). Through electrospinning, the components of each half of the basal membrane between the two types of cells are made to be able to be doped with antibodies specific to the contralateral matrix. These antibodies may be whole, leading to a high efficiency of the contralateral matrix fixation process, or they may only be formed from the Fab region to prevent recognition by the immune system with the development of self-immunity.

The main limitation of such an approach is the commercial cost associated with it, which mainly encompasses the very high research–development costs involved compared to the relatively small number of organs needed for transplantation. This stage of research–development, which involves a relatively small number of cells, is imperative to understand the variables described as the intermediate stage in the realization of artificial organs with a high number of cells, such as in the case of the kidney. In this case, there are extremely complex signaling mechanisms at the level of ion channels [70,71,72,73,74,75,76,77,78,79,80,81,82,83,84,85,86,87] (which must be individually identified at the initial stage) and ion pumps, which require a multitude of cell types.

The current limitations of 3D bioprinting based on spheroids may be secondary to the metabolic inefficiency of the printed organ, especially in the case of organs with a complex metabolism such as the liver. In this sense, it is interesting to observe how biological evolution has managed to increase efficiency through processes of compartmentalization at the cellular level and supracellular organization which modulates the intracellular organization. Although the interactions between the extracellular matrix and cells have been shown to have a major role in cell orientation, including metabolic orientation, spheroid-type simplifying models have continued to be the central element in the 3D bioprinting industry. This approach is secondary to the need for a high printing speed to cover the acute needs of patients and to the need for more accessible modern technologies, which, as stated earlier, are becoming more and more complex and less economically accessible.

The ECM is produced by cells under simplified culture conditions so that cells can predictably communicate with it according to their genetic codes. Reduced ECM turnover preserves the architecture of the 3D-printed organ (including at the intracellular level) in the printing stages that are not similar to the physiological process of organ formation. After printing the designed structures, communication with the environment is resumed (for example, through blood and bile flows). In this case, the basement membranes in the 2D cell culture function as a slow extracellular memory. The specialized cells “write” the appropriate information in this “memory”, which the cells later use for “reading” when all communication with the environment is inconsistent with the evolutionary model of organogenesis (in the 3D printing process).

## 8. Limitations of the Proposed Model

The high degree of multi-disciplinarity, ranging from elements of cellular and molecular biology to elements of material science (for the realization of the cellular spot) and continuing with elements of microfluidic, electronic, and mathematical modeling (for the realization of the tissue printer), necessitates a strong interdisciplinary team with multiple feedback loops between them. For example, rotating the cellular spot in the electric fieldrequires the development of materials with a high degree of electrical charge according to small differences in pH (within physiological limits), the optical fluorescent assessment of the rotational positioning (possibly with a fluorophore at the level of the support material and another at the cellular level), flow cytometer sorting starting from more than one channel (compared to classical systems), and rotation using a high number of electrodes (compared to a minimum of four in the flow cytometer).

However, an overall plan of a possible technological solution that attracts large and diverse teams of researchers is not enough; appropriate funding, which does not seem to be easily obtainable in the context of using artificial organs, is also needed. This assessment is based on the relatively small number of commercially available bioprinting technologies compared to the number of articles showing experimental results or theoretical models. Ideally, if an economic profit is pursued through a patented technology, the publication of results can only be carried out after ensuring the protection of the technology through a patent application. This suggests that many authors in scientific journals do not estimate that their financial effect is unlikely to yield a commercial technological approach, and consequently, they remain in the field of academic publications.

In order to attract financially powerful stakeholders over a significant period of time, it is necessary to identify diversified ways of recovering invested resources, starting from direct financial ones (from the implementation of the biomedical part, which is relatively slow due to legal regulations and the size of market demand) and continuing with collateral financial effects (in non-medical applications) to not overlooking the image effects that can strengthen the market effect of such a financier.

The many legal regulations [88] in the medical field increase the difficulty of developing a new technology while also increasing the cost. These regulations must be identified and followed from the initial stages of the design process. The regulations, combined with the relatively small number of users that could benefit from this technology, will decrease the profits that can be obtained from this otherwise very complex and hard-to-design product. To make this product viable on the market, another related use for it must be found until approval for medical applications is obtained.

## 9. Possible Non-Medical Applications

If we could make an artificial liver through using a theoretical model that is not based on organo-genetic evolution in intrauterine life, we could theoretically use genetically modified cells relatively easily (compared to the development of a model that includes the integration of these changes in the onto-genetic evolution), such as those with new channels or pumps and those that have been adapted to take on new roles. Research on ion channels and pumps is highly prolific (due to their pharmaceutical implications) and concerned with understanding signaling and regulation mechanisms through molecular simulation and the design of proteins with new roles.

A possible non-medical application of molecular technologies is the extraction of polluting substances using proteins of biological origin (usually bacterial). An application that could increase the impact of de novo molecular biology via a combination with cellular anisotropy is the ability to pump and/or fix various components of interest. Making an artificial liver in a relatively autonomous animal (e.g., sheep) that is capable of sorting, concentrating, and/or fixing physiologically non-usual elements would have a wide range of applications. Among these potential applications could be the decontamination of pollutive or rare substances on large surfaces [e.g., radioactive substances, polluting substances with commercial use (lithium), rare earth elements] via extracting them from the environment. After the first bacterial concentration stage of the targeted substances, a second concentration stage could use complex transport mechanisms from the liver of genetically modified animals. Such separation and concentration in two biological steps can overcome the initial limitations of bacteria separation.

The key difference between biological systems and contemporary technologies [89] lies within the energy efficiency of the biological systems that are used for industrial purposes. Converting traditional mining areas into pollution-free areas can be achieved using biotechnologies that allow for remediation [90,91] and economic profit [92] simultaneously from the extraction of useful and/or dangerous substances [93,94]. We must not forget that one of the reasons for the slow development of lithium-based batteries in modern cars is the limited resources of lithium, including the poor technologies employed to recover it from used products and its association with increased brain toxicity.

The proposed technology meets the highest sustainability standards; it could help to facilitate a significant decrease in pollution and our dependence on important non-renewable energy resources because it is energy efficient and, since it is based on a biological material, entirely biodegradable.

## 10. Conclusions

The theoretical model presented by 3D tissue printing preserves cellular asymmetry and is based on current technologies; these two aspects are combined in a manner that ensures function. Starting from the creation of cellular spots consisting of at least two cells to ensure cellular asymmetry, 3D complex pieces can be obtained. The mechanisms of mechanoreceptors at integrin and cadherin levels were considered to be the main mechanisms for ensuring cellular asymmetry. Starting from the asymmetric distribution of these two types of adhesion molecules, a theoretical model for preserving this distribution in the 3D printing process was developed; it will be subsequently tested to find out whether the other signaling pathways are reconstituted at the level of the 3D printed tissue through having a simple cellular proximity similar to the targeted organ.

The need for printing during the infusion of the 3D constructed piece was, through making reference to the technological reasons associated with the printing process, highlighted as a simplifying element of the orientation of the cellular spot in the electric field. The use of electrically charged substances with an isoelectric pH in the physiological margin allows for the application of rotational forces during 3D printing but with the loss of this electrical polarization when reaching the 3D construction, respectively against the electric fields in the printing device.

The novelty of our 3D bioprinting technology model lies within its use of oriented multicellular blocks and the fact that its process ensures fast printing speeds and the ease of obtaining the printing blocks. The preservation of the structures obtained in 2D cell culture through the 3D bioprinting process is based on the low turnover of the extracellular matrix compared to the speed of the modulation of intracellular activities.

This article suggests multiple design solutions for the issue of 3D printing complex tissue, emphasizing the advantages and disadvantages of each of them. In order to simplify the theoretical model and the implementation of the first 3D-printed tissue with cellular asymmetry, potential medical (liver tissue) and non-medical applications (extracting pollutants) were discussed.

## Figures and Tables

**Figure 1 ijms-24-14722-f001:**
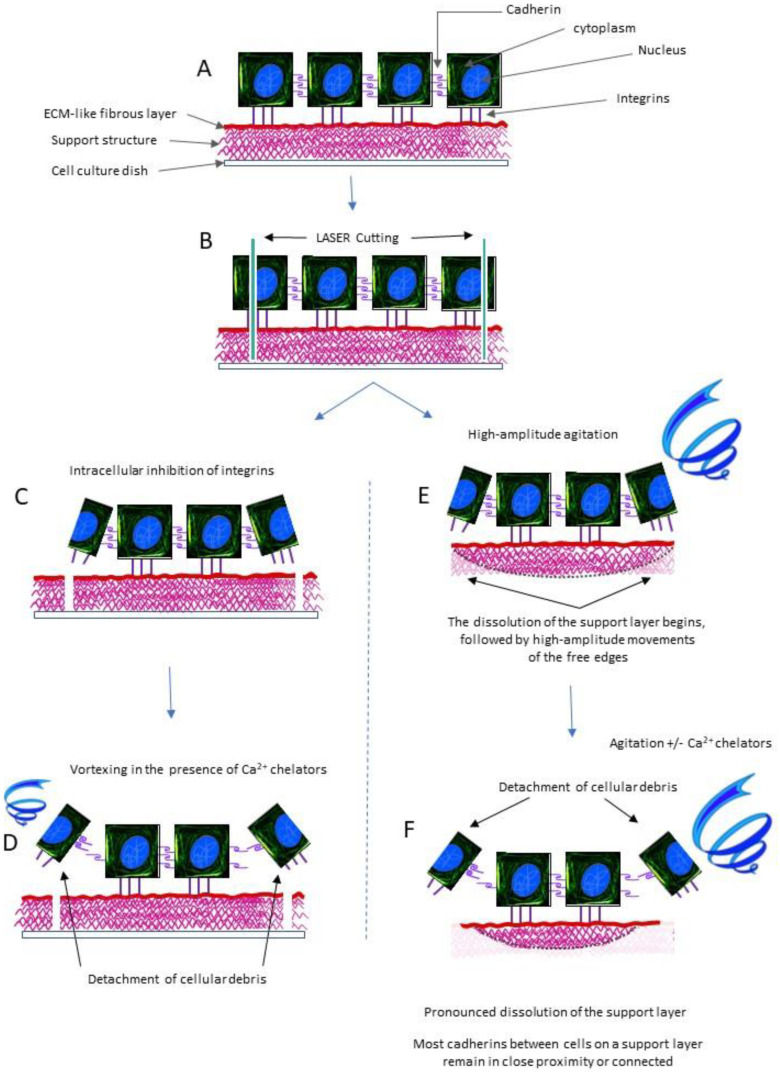
Cellular spot production process. (**A**) Cell culture on support matrix. (**B**) LASER cutting. (**C**) Substrate detachment of integrins from cell fragments. (**D**) Separation of cellular debris via vortexing and the use of calcium chelators. (**E**) Substrate detachment of integrins from cell fragments at controlled mechanical instability at edges of support matrix induced by vortexing and the dissolution of the support layer. (**F**) Detachment of cellular debris induced by increasing the dissolution of the support layer and using calcium chelators.

**Figure 2 ijms-24-14722-f002:**
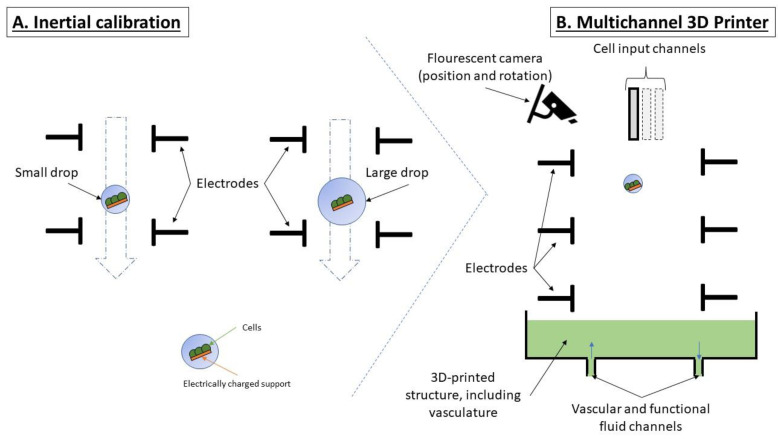
The 3D printing of cellular tissue. (**A**) Inertial calibration of cellular spot; (**B**) 3D tissue printing using multiple cell types.

**Table 1 ijms-24-14722-t001:** Characteristics of different bioprinting set-ups in relation to cell asymmetry.

Bioprinting Technology	Printing Resolution	Sequential Printing	Toxic Elements
Extrusion bioprinting	Down to 1 μm	No	Some systems use UV light exposure
Inkjet bioprinting	Down to 25 μm	Yes	Thermal stress
			Shear stress
Laser-assisted bioprinting	High	Yes	Yes
Photocuring-based bioprinting	Down to 5 μm	No	UV light exposure

## Data Availability

Not applicable.

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
