# Peer review of "Perspectives on Scaffold Designs with Roles in Liver Cell Asymmetry and Medical and Industrial Applications by Using a New Type of Specialized 3D Bioprinter"

_ijms, 2023, doi:10.3390/ijms241914722_

Round 1
Reviewer 1 Report (Previous Reviewer 2)
The authors have addressed all the queries. Recommended for Publication
Author Response
We thank the reviewer for the positive assessment.
Reviewer 2 Report (New Reviewer)
Authors have presented a significant study entitled "Scaffold design perspectives with role in liver cell asymmetry with medical and industrial applications". However, it needs extensive editing.
The title needs to be modified as it looks very simple. Please reframe the title to increase its significance.
The abstract should present the novelty of the study hence it needs to get reframed.
The introduction section is very poorly organized. More citations are needed to increase the significance of the study.
The discussion section is very weak.
The conclusion needs to be improved.
Increase the number of citations and incorporate more recent references.
English needs to be improved.
Author Response
The title has been modified to be more comprehensive.
In the abstract, the novelty of the theoretical model was emphasized.
The introduction section contains several new citations and has been partially reorganized.
The discussions and conclusions also contain an explanation of how the different rates of change in the intracellular environment and the ECM are the key to maintaining cellular anisotropy in the printing process.
The number of cited articles from recent years has been increased.
Reviewer 3 Report (New Reviewer)
The current article titled “Scaffold design perspectives with role in liver cell asymmetry 2 with medical and industrial applications” Ref: ijms-2488408, deals with an important recent subject. It is a useful perspective giving overview information about the titled subject. Minor revision is needed.
A short paragraph describing the previous publications/reviews/perspectives about the titled subject with comprehensive discussion may enhance the impact of the article.
Author Response
The number of citations of recent technological developments has increased.
Discussions were developed to explain how different turnover rates for the intracellular environment and the ECM can maintain cellular asymmetry.
Reviewer 4 Report (New Reviewer)
The article provides interesting information in the field of scaffolding for liver asymmetry however, the Authors should pay more attention to innovative and new approaches in this topic. Subsequently, the development perspective in recent times should be highlighted more. Including what has changed in the past time what is the progress and further direction of development.
Author Response
Highlights of the innovative elements of recent years have been introduced.
Among the new elements appearing in the 3D printing of organs, the authors selected only the elements considered relevant for an easy understanding of the proposed theoretical model. In this sense, the authors avoided as much as possible the interpolation of data from the literature with the proposed model, separating the levels as precisely as possible to make the level of novelty evident.
The relatively brief presentation of the evolution of bioprinting is not disrespectful to the undeniable effort of researchers over time, but is a method of increasing the accessibility of the article to researchers in the traditional printing industry, not in the bioprinting industry (where these developments are well known). This step of highlighting critical elements with an apparent disregard for those deemed non-essential is a method of developing simplifying models that are easier to test in an interdisciplinary manner.
Being a theoretical model, the authors tried to predict the next generation of organ printers, respectively indicated a possible further evolution. Moreover, the authors tried to identify non-medical uses of the proposed printer model.
Round 2
Reviewer 2 Report (New Reviewer)
Authors have incorporated the suggested changes and hence can be accepted.
Authors have incorporated the suggested changes and hence can be accepted.
This manuscript is a resubmission of an earlier submission. The following is a list of the peer review reports and author responses from that submission.
Round 1
Reviewer 1 Report
The authors try to establish a theoretical model for preserving cellular asymmetry during 3D tissue printing. However, their hypothesis is based on an immature technique with no much applications in organ, especially liver 3D printing. For the actual 3D bioprinting, the opinions and statements of the authors are all extremely partial, lacking of universal meanings and practical values. This reviewer suggests to reject it without further processing.
Reviewer 2 Report
The research article entitled “Scaffold design perspectives with a role in liver cell asymmetry modeling” discussed the application of 3D printing in liver tissue engineering. The authors have also highlighted the complex structure of live and cell asymmetry modeling. However, the manuscript should be further upgraded by adding a brief about the various methods of 3D printing with their advantages and disadvantages, Commercialization and regulatory aspects, etc. The Review article entitled “ Under this concern and the comments below, I suggest this paper could be published in pharmaceutics after minor revision.
1. Title does not reflect the various data provided in the manuscript like 3D printing
2. Authors need to discuss various methods of 3D printing with their advantages and disadvantages in a separate section. Refer https://doi.org/10.1016/j.bprint.2022.e00208
3. comprehensive Information in the form of a table regarding the area's clinical studies needs to be provided.
4. Commercialization and regulatory aspects of 3D printing-based scaffolds for liver tissue engineering should be included.
5. Authors need to improve the references by citing recent references like
https://doi.org/10.1016/j.trsl.2019.04.002
https://doi.org/10.1016/B978-0-12-824064-9.00017-4
https://doi.org/10.1016/j.bprint.2022.e00208
6. Typographic errors must be corrected (Example: line 239). The language and grammar used throughout the manuscript need to be improved
